# Rapid translocation of NGR proteins driving polarization of PIN-activating D6 protein kinase during root gravitropism

Ivan Kulich[†], Julia Schmid[†], Anastasia Teplova[†], Linlin Qi, Jiří Friml*

Institute of Science and Technology Austria, Klosterneuburg, Austria

*For correspondence:
jiri.friml@ist.ac.at

[†]These authors contributed equally to this work

Competing interest: The authors declare that no competing interests exist.

**Abstract** Root gravitropic bending represents a fundamental aspect of terrestrial plant physiology. Gravity is perceived by sedimentation of starch-rich plastids (statoliths) to the bottom of the central root cap cells. Following gravity perception, intercellular auxin transport is redirected downwards leading to an asymmetric auxin accumulation at the lower root side causing inhibition of cell expansion, ultimately resulting in downwards bending. How gravity-induced statoliths repositioning is translated into asymmetric auxin distribution remains unclear despite PIN auxin efflux carriers and the Negative Gravitropic Response of roots (NGR) proteins polarize along statolith sedimentation, thus providing a plausible mechanism for auxin flow redirection. In this study, using a functional NGR1-GFP construct, we visualized the NGR1 localization on the statolith surface and plasma membrane (PM) domains in close proximity to the statoliths, correlating with their movements. We determined that NGR1 binding to these PM domains is indispensable for NGR1 functionality and relies on cysteine acylation and adjacent polybasic regions as well as on lipid and sterol PM composition. Detailed timing of the early events following graviperception suggested that both NGR1 repolarization and initial auxin asymmetry precede the visible PIN3 polarization. This discrepancy motivated us to unveil a rapid, NGR-dependent translocation of PIN-activating AGCVIII kinase D6PK towards lower PMs of gravity-perceiving cells, thus providing an attractive model for rapid redirection of auxin fluxes following gravistimulation.

## eLife assessment

This **fundamental** study addresses the earliest events that enable plant roots to reorient growth in response to gravity. **Compelling** molecular and cell biological data establish that plasma membrane localization of the LAZY or NEGATIVE GRAVITROPIC RESPONSE OF ROOTS (NGR) protein family is required for rapid and polar redirection of D6 protein kinase, an activator of the PIN3 auxin transporter. This work complements and extends recent publications on the NGR family in gravity sensing (PMID: 37741279 and PMID: 37561884). Collectively these papers advance our understanding of rapid plant gravity sensing and response.

## Introduction

Plants are capable of sensing gravity and adjust their organs' growth direction accordingly. This phenomenon is called gravitropism, with positive gravitropism referring to the root growing down towards the Earth center and negative gravitropism describing the upward growth of the shoot (*Kawamoto and Morita, 2022*; *Kolesnikov et al., 2016*; *Takahashi et al., 2021*). The model plant *Arabidopsis thaliana* senses gravity in central root cap (columella) cells and the shoot endodermal cells and, in special gravity-sensing cells, which contain starch-enriched plastids – amyloplasts (*Kiss et al., 1997*; *Kiss et al., 1996*; *Kiss et al., 1989*; *Morita, 2010*). Due to their higher density, amyloplasts thus

act as statoliths and sediment within cells (*Sack, 1991*). Following amyloplast sedimentation, the PIN exporters for the plant hormone auxin (PIN3 and PIN7) polarize towards the bottom cell sides, thus creating a downwards polarized auxin flow, which correlates with the asymmetric auxin accumulation at the lower side of the organ mediating the shoot/root curvature (*Friml et al., 2002*; *Kleine-Vehn et al., 2010*; *Rakusová et al., 2011*; *Band et al., 2012*). While sedimenting amyloplasts clearly serve as a signal for asymmetric auxin distribution, the precise mechanism remains elusive (*Han et al., 2021*; *Kiss et al., 1989*).

Recently, the LAZY protein family has been identified as a crucial player in the process of auxin redistribution and normal tropic response for both roots and shoots following gravistimulation (*Jiao et al., 2021*; *Li et al., 2007*; *Yoshihara et al., 2013*; *Yoshihara and Iino, 2007*). LZY1, LZY2, and LZY3 function redundantly in shoot gravitropism; LZY2, LZY3, and LZY4 - also referred to as NGR1, NGR3, and NGR2 (NEGATIVE GRAVITROPIC RESPONSE OF ROOTS) - regulate the root gravitropism (*Kawamoto and Morita, 2022*). *ngr1,2,3* triple mutant plants display an anti-gravitropic phenotype with the root growing slightly upwards (*Ge and Chen, 2016*). This phenotype is accompanied by an inverted PIN3 polarization, thus redirecting the auxin flow to the upper side of the gravistimulated root (*Ge and Chen, 2019*; *Taniguchi et al., 2017*). Additionally, it was recently demonstrated that following gravistimulation, NGR3 localizes asymmetrically to the plasma membrane (PM) in columella cells of lateral roots (*Furutani et al., 2020*). However, the structure and molecular function of NGR proteins, as well as specific domains that could allow for PM association and polarization are still unknown. Polarization of NGR3 further acts as a precursor of PIN3 relocalization (*Furutani et al., 2020*). Additionally, PIN transport activity is fine-tuned by phosphorylation mediated by AGCVIII serine/threonine kinases like D6 PROTEIN KINASE (D6PK) (*Barbosa and Schwechheimer, 2014*). Whether additional factors play a role in this signaling cascade and how the underlying molecular mechanism connecting gravity sensing to asymmetric auxin distribution may look like, remains to be investigated.

Here, we provide novel insights into mechanisms of gravity-induced NGR1 polarization and downstream processes leading to PIN-dependent auxin fluxes activation and ultimately, gravitropic bending. We show the crucial function of polybasic regions (PBRs) and acylation of NGR1 for its binding to the PM and amyloplasts in columella cells, which is essential for its function in gravitropic bending. Pharmacological approaches show that the membrane composition affects NGR1 PM binding and polarization. Finally, we show rapid translocation of PIN activator – D6 protein kinase as an event downstream of NGR signaling. These observations suggest a model where statolith sedimentation is accompanied by rapid NGR relocation, which mediates similar relocation of the D6PK activator of PIN transporters, providing means to rapidly redirect asymmetric auxin fluxes in response to gravity.

## Results

### NGR1 localizes to statoliths and their sedimentation is accompanied by NGR1 polarization at the lower PM

To observe NGR localization in the columella cells, we cloned *NGR1-GFP* under its native promoter (*NGR1p::NGR1-GFP*) and transformed it into the *ngr1/2/3* triple mutants. This construct fully rescued the agravitropic bending phenotype of the *ngr1/2/3* triple mutant (see further), proving functionality of the GFP-tagged protein. NGR1-GFP displayed a highly specific columella expression, which was most prominent at the PM and the statolith periphery. NGR1-GFP signal from the PM was not evenly distributed, but rather polarized to the lower side of the columella cells in the vicinity of the sedimented statoliths (*Figure 1A*).

Constitutive overexpression of NGR1-GFP under 35 S promoter (*35Sp::NGR1-GFP*) confirmed that in multiple other cell types, NGR1-GFP localizes to the PM suggesting direct membrane binding mechanism. Additionally, NGR1 is associated with other plastid types such as chloroplasts in guard cells (*Figure 1—figure supplement 1A*). To separate the GFP signal from the chlorophyll, we used fluorescence lifetime imaging, which confirmed that NGR1-GFP localizes at the chloroplast periphery (*Figure 1—figure supplement 1B*). This implies a general plastid targeting mechanism of NGR. Notably, chloroplasts in the vicinity of the PM strongly correlated with NGR1 accumulating at the PM nearby, similar to the scenario in columella (*Figure 1—figure supplement 1A and C*).

This shows that NGR1-GFP localizes to the plastids and PM in their vicinity providing a mechanism for how gravity-induced statolith sedimentation can polarize NGR1-GFP to the bottom PM.

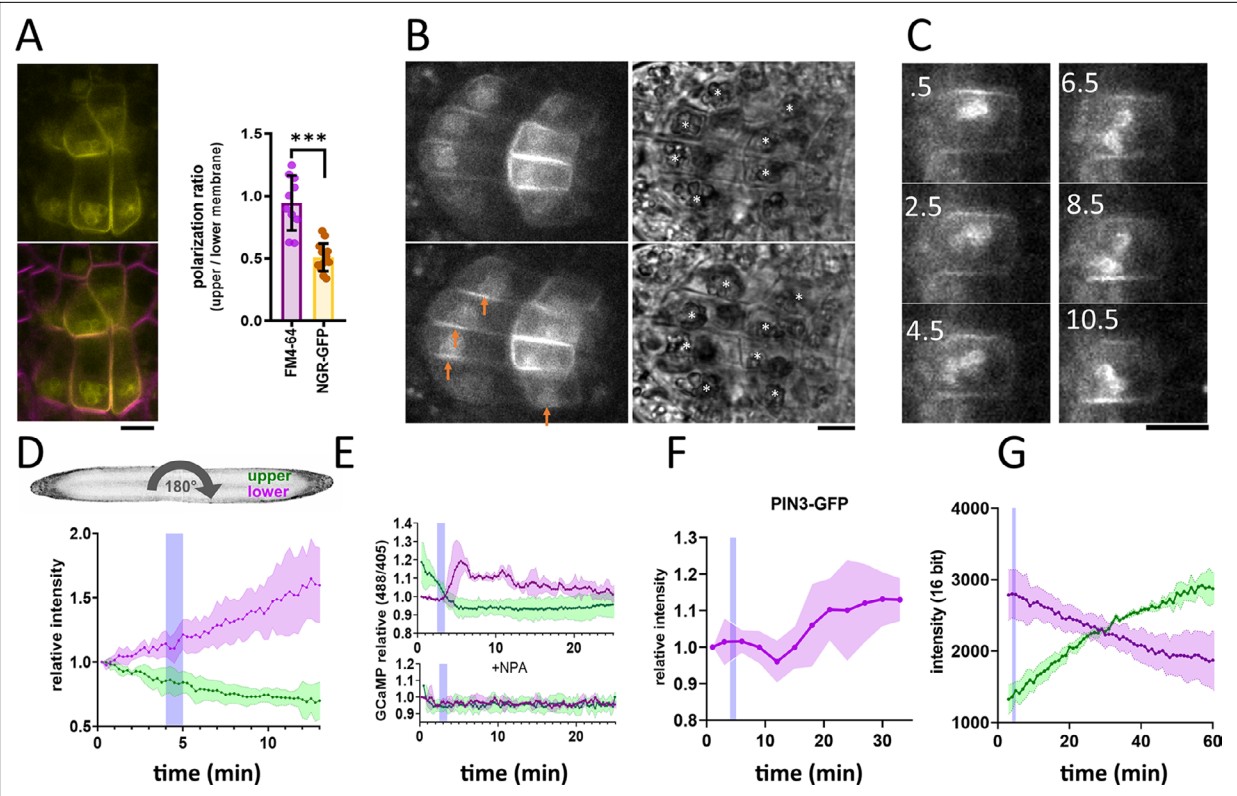

**Figure 1.** NGR1-GFP relocalization in the root columella. (**A**) Colocalization of NGR1-GFP (yellow) with membrane stain FM4-64 (magenta). The statolith periphery is also decorated by the NGR1-GFP signal. NGR1-GFP is significantly polarized to the bottom area as quantified using upper/lower signal ratio (1=even distribution). (**B**) NGR1-GFP~30 s and 5 min after gravitropic bending. Orange arrows depict areas where the signal increased after contact with the statoliths. White stars in the transmission channel depict statoliths. (**C**) Time course of individual columella cell expressing NGR1-GFP. Numbers depict minutes from the 180° rotation. (**D**) Top: schematic depiction of the experiment design (valid for D-G). Bottom: Quantification of the NGR1-GFP signal at the lower and upper membranes. Similar setup was used for E. and F. Blue stripes in D-F represent intervals at which statoliths touched the bottom membrane. Average of five biological replicas. Shaded regions depict standard deviation (SD). (**E**) GCaMP signal after root gravitropic bending without (top) and with 20 µM NPA (bottom). The data represent three technical replicates, and three additional biological replicates were conducted, yielding similar results. Shaded regions depict SD.(**F**) PIN3 enrichment at the bottom of the central columella cells. The average is based on data from five biological replicas. Shaded regions depict SD.(**G**) DII Venus signal changes after root gravitropic bending. The average is calculated from three biological replicas. Shaded regions depict SD. Blue lines represent approximate time of statolith contact with the new bottom membrane. Bars = 10 µm.

The online version of this article includes the following source data and figure supplement(s) for figure 1:

**Source data 1.** Data used for generating the graphs in the figure.

**Figure supplement 1.** NGR1 localization in columella and guard cells.

**Figure supplement 1—source data 1.** Data used for generating the graphs in the figure.

## Timing of events during the gravitropic response

Next, we tracked the timing of events during gravitropic bending. For that, we used a vertical stage microscope with a rotation stage allowing us to perform gravistimulation while imaging and minimize the time required for rotating the sample. For the gravistimulation, roots, initially located vertically, were rotated 90° to the horizontal position.Further gravistimulation was performed by flipping the root 180° (with the root tip from pointing left to pointing right, or vice versa). 180° rotations were performed during the imaging thus the precise timing could be determined from the individual frame timestamps. Following gravitropic stimulus, NGR1 at the PM relocalized dynamically with the falling statoliths (*Figure 1B*). First, the signal at the top membrane decreased and then reached the bottom PM together with statoliths, around 5 min after the stimulation (*Figure 1C and D*). If the root was rotated 180° from the vertical position (from root tip pointing down to pointing up), the time required for the statolith sedimentation has increased due to the cell's elongated shape and the presence of vacuole. In such a scenario, it took more than 15 min for the statoliths to fall. Again, the increase

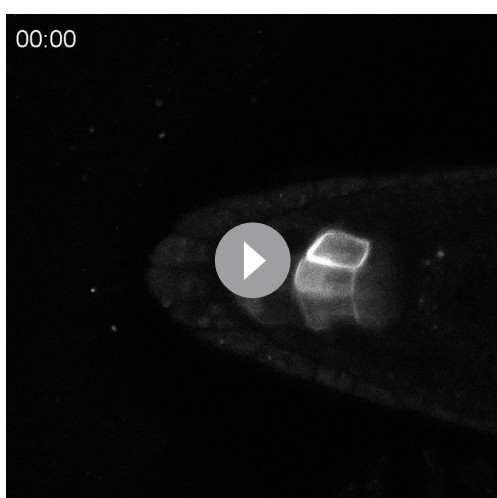

00:00

**Video 1.** NGR-GFP protein localizes in the vicinity of statoliths during the course of root gravitropic bending. https://elifesciences.org/articles/91523/figures#video1

of the NGR1 signal at the bottom side correlated with the statolith occurrence, implying that statolith vicinity is the region where NGR1 signal gets enriched (*Figure 1—figure supplement 1D*). When monitored for an extended period of time, NGR1 signal was always decorating the PM proximal to the statolith (*Video 1*).

To put these early gravitropic events in the context of the auxin gradient formation, we tracked PIN3 localization dynamics, intracellular calcium transients, and auxin reporter DII-Venus (*Brunoud et al., 2012*). Calcium transients are one of the fastest known auxin response events (*Monshausen et al., 2011*) and we monitored them using plants stably transformed with GCaMP3 fluorescent sensor (*Zariwala et al., 2012*). While the signal increase at the bottom side of the root was visible within a minute after statoliths touched the bottom membrane, its decrease at the top PM started immediately after rotation. This implies that the first signaling event in the root gravitropic bending is the statolith removal from the top membrane, rather than its arrival at the bottom (*Figure 1E*). While subsequent changes in the DII-Venus degradation were consistent with the calcium-based timing, the first visible PIN3 polarization to the bottom columella cell sides was detectable only after 15 minutes (*Figure 1F and G*). Nonetheless, already the earliest auxin asymmetry is likely generated by PIN-mediated polar auxin transport as it was completely blocked by the treatment with N-1-naphthylphthalamic acid (NPA) (*Figure 1E*), a PIN inhibitor (*Abas et al., 2021*).

To summarize, statolith sedimentation is accompanied by rapid NGR1 relocation to the PM subdomain proximal to the NGR1-containing statoliths. The first event in gravitropic response is the NGR1 leaving the top membrane accompanied by decrease in the auxin signaling at the upper root side, followed by NGR1 arrival at the bottom membrane and increase in auxin signaling at the bottom side. This initial auxin asymmetry is mediated by PIN-dependent auxin transport, despite visible polarization of PIN3 can be detected only later.

## NGR1 polarization does not require BFA-sensitive trafficking

To gain more insight into the dynamics of the NGR1 localization to the PM and amyloplasts, we tested different inhibitors to alter membrane composition and interfere with endocytic trafficking. Recently, it has been shown that RCC1-like domain (RLD) proteins directly interact with NGRs and play a crucial role in polar auxin transport during gravity signaling (*Furutani et al., 2020*). As RLD trafficking is known to be sensitive to the ARF-GEF inhibitor Brefeldin A (BFA) (*Furutani et al., 2020*; *Wang et al., 2022*), we examined if the drug also affects the localization and polarization of NGR1. Five-day-old *NGR1p::NGR1-GFP* seedlings were incubated with 50 µM BFA together with the protein synthesis inhibitor, cycloheximide (CHX, 50 µM) and the endocytic tracer FM4-64 (2 µM) as described previously (*Geldner et al., 2001*). Interestingly, while FM4-64-stained membranes accumulated into so-called BFA compartments, NGR1-GFP did not and still showed association with the PM and amyloplasts (*Figure 2A–C*). Furthermore, the polarization of NGR1 following gravity stimulation was not disrupted in the presence of BFA. Similar effects were observed in overexpressed NGR1 as it also did not show co-localization with BFA aggregates (*Figure 2—figure supplement 1A–C*).

This shows that NGR1 localization and gravity-induced polarization does not undergo BFA-sensitive endocytic recycling mediated by the ARF-GEF GNOM as it has been shown for other proteins involved in gravity response like RLDs and PINs (*Geldner et al., 2001*; *Naramoto et al., 2014*; *Wang et al., 2022*).

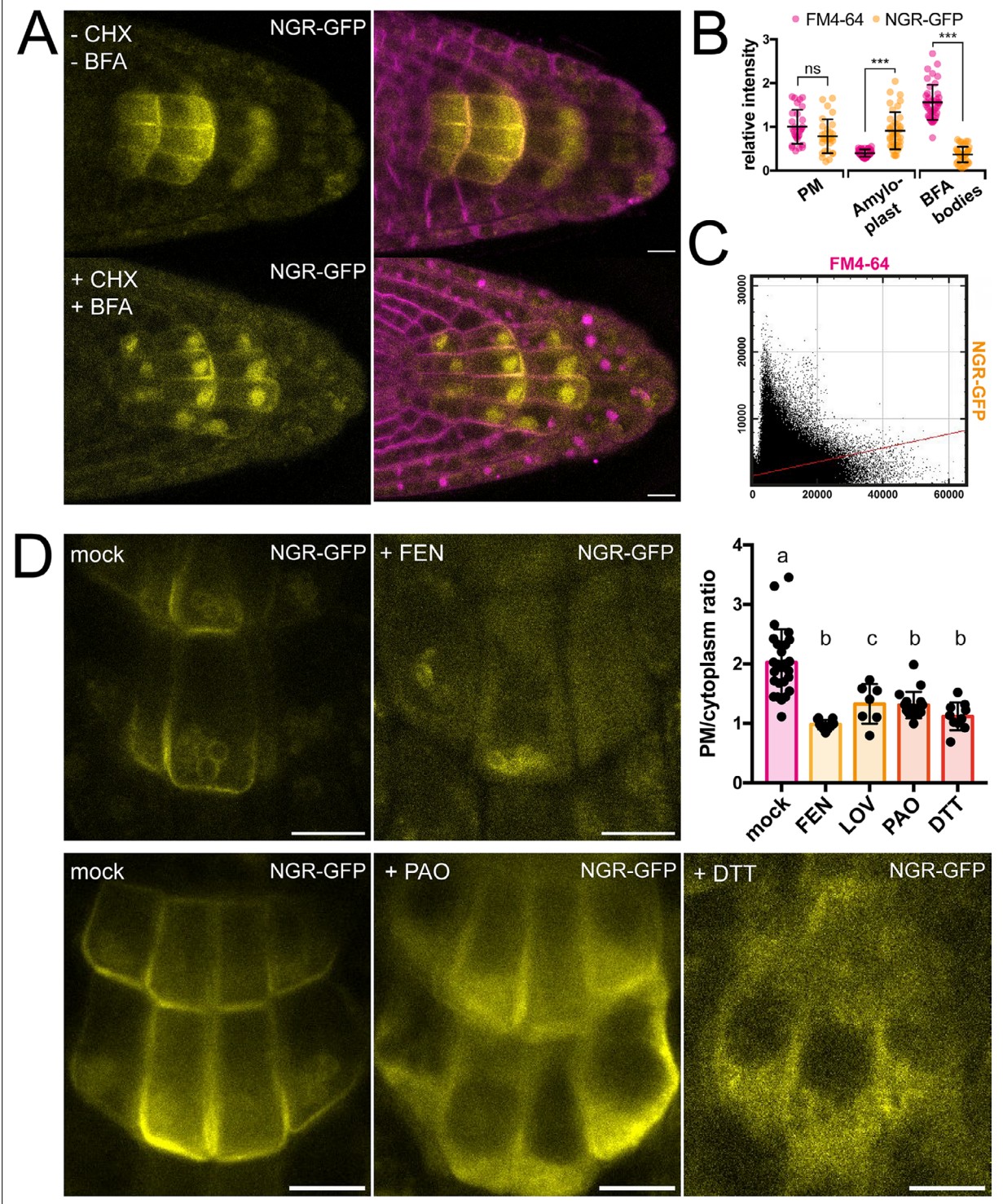

**Figure 2.** Role of trafficking and PM composition in NGR1 localization. (**A**) Representative images of *NGR1p::NGR1-GFP* roots treated with 50 µM CHX, 50 µM BFA, and 2 µM FM4-64; mock: DMSO; NGR1-GFP in yellow, FM4-64 in magenta. Three biological replicas were made with similar results. Error bars represent SD. (**B**) Fluorescence intensity of FM4-64 and NGR1-GFP at the PM (n=29), amyloplasts (n=39) and BFA bodies (n=42) normalized to the average fluorescence intensity of FM4-64 at the PM. Data is pooled from columella cells of the representative experiment shown in **A**. *n* equals measured points. Statistical measurements were done with one-way ANOVA; *** equals adj. p-value of <0.0001. (**C**) Cytofluorogram of colocalization of FM4-64 and NGR1-GFP from the **A** using Pearson's coefficient. (**D**) *NGR1p::NGR1-GFP* roots treated with FEN, PAO, and DTT. PM/cytoplasm ratio was calculated for mock (n=26), FEN (n=11), LOV (n=7), PAO (n=18), and DTT (n=10) treated seedlings. *n* equals the number of cells evaluated. Using

*Figure 2 continued on next page*

*Figure 2 continued*

one-way ANOVA, *b* is significantly different from *a* with an adj. p-value of <0.0001, *c* is significantly different from *a* with an adj. p-value of 0.006. Bars = 10 µm.

The online version of this article includes the following source data and figure supplement(s) for figure 2:

**Source data 1.** Data used for generating the graphs in the figure.

**Figure supplement 1.** Pharmacological treatments of *35Sp::NGR1-GFP* seedlings.

**Figure supplement 1—source data 1.** Data used for generating the graphs in the figure.

## NGR1 PM association requires specific membrane composition

NGR1 displays PM localization although transmembrane domains are not predicted in its structure and it does not undergo BFA-sensitive trafficking. This implies the existence of posttranslational modifications such as S-acylation or N-myristoylation to associate with PM. For initial insight we used GPS Lipid prediction tool (*Xie et al., 2016*), which navigated us towards S-acylation. To test for the presence of S-acylation sites, we utilized the reducing properties of Dithiothreitol (DTT), which interferes with the formation of the thioester bond thus effectively preventing S-acylation (*Sabol et al., 2017*). Following previously described conditions, NGR1-GFP showed an increase in cytoplasmic signal and reduction of PM association (*Figure 2D*). Similar effect of DTT was observed when expressing NGR1 under the 35 S promoter, suggesting that there is indeed an acylation site (*Figure 2—figure supplement 1D–E*). Interestingly, NGR1 still localizes to amyloplasts, indicating a different way of plastid association compared to the PM.

Next, we tested whether apart from the NGR1 S-acylation, the membrane composition itself might be essential for the NGR1 PM association. Previously, it has been shown that PBRs of proteins can target them to anionic phospholipids like PtdIns(4)P (PI4P) (*Hammond et al., 2012*; *McLaughlin and Murray, 2005*). We tested whether the depletion of PI4P from the PM will affect the NGR1 PM association. *NGR1p::NGR1-GFP* seedlings were incubated for 30 min with 30 µM phenylarsine oxide (PAO), a PtdIns(4)-kinase inhibitor (*Hammond et al., 2012*; *Vermeer et al., 2009*). The NGR1-GFP cytosolic signal was significantly increased, indicating that the interplay between PBRs and phospholipids is important for NGR1 PM but not amyloplast association (*Figure 2D*).

Next, we wanted to determine whether sterols might influence the PM association of NGR1. *NGR1-GFP* seedlings were grown on fenpropimorph (FEN), an inhibitor of sterol biosynthesis that decreases the number of sterols and can trigger their conversion into cyclopropyl sterols (*Frescatada-Rosa et al., 2014*). NGR1-GFP showed an increase in cytoplasmic signal and membrane localization was largely lost (*Figure 2D*). Comparable effects were also observed after treatment with another sterol synthesis inhibitor lovastatin (LOV) (*Figure 2D*; *Figure 2—figure supplement 1F*). However, similarly to DTT and PAO, amyloplast localization could not be disturbed.

These observations show that NGR1 PM but not necessarily plastid localization depends likely on acylation, phospholipid, and sterol membrane composition.

## NGR1 PM localization is synergistically mediated by PBRs and a palmitoylation site

To further study NGR1 membrane localization, we examined the amino acid sequence of the NGR1 to understand which sites of the protein mediate its association with the membrane. We noticed 2 PBRs $K^{113}K^{114}R^{115}K^{116}$ (PBR1) and $K^{186}K^{187}K^{188}R^{189}$ (PBR2) – these strands of positively charged amino acids may mediate the interaction of NGR1 with negatively charged phospholipids. Also, we noticed a conserved $Cys^{206}$ residue – a potential S-palmitoylation (PALM) site according to the GPS-Lipid posttranslational modification prediction tool (*Xie et al., 2016*).

To test the role of these sites in the NGR1 membrane localization we performed site-directed mutagenesis based on the *NGR1p::NGR1-GFP* construct. We introduced mutations $K^{114}E\&R^{115}E$ (-PBR1), and $K^{187}E\&K^{188}E$ (-PBR2) to the PBRs of the NGR1 to neutralize their positive charge, and mutated the conserved cysteine $C^{206}A$ (-PALM) to abolish the potential palmitoylation site (*Figure 3A*). *NGR1-PALM-GFP*, *NGR1-PBR2-GFP*, and *NGR1-PBR1&2-GFP* under the native *NGR1* promoter were transformed into the *ngr1/2/3* background. These mutant versions of the *NGR1* restored the anti-gravitropic root phenotype of the *ngr1/2/3* triple mutant similar to the *NGR1-GFP* version without mutations (*Figure 3—figure supplement 1A*). The *NGR1-PALM* version resembled the root bending of the

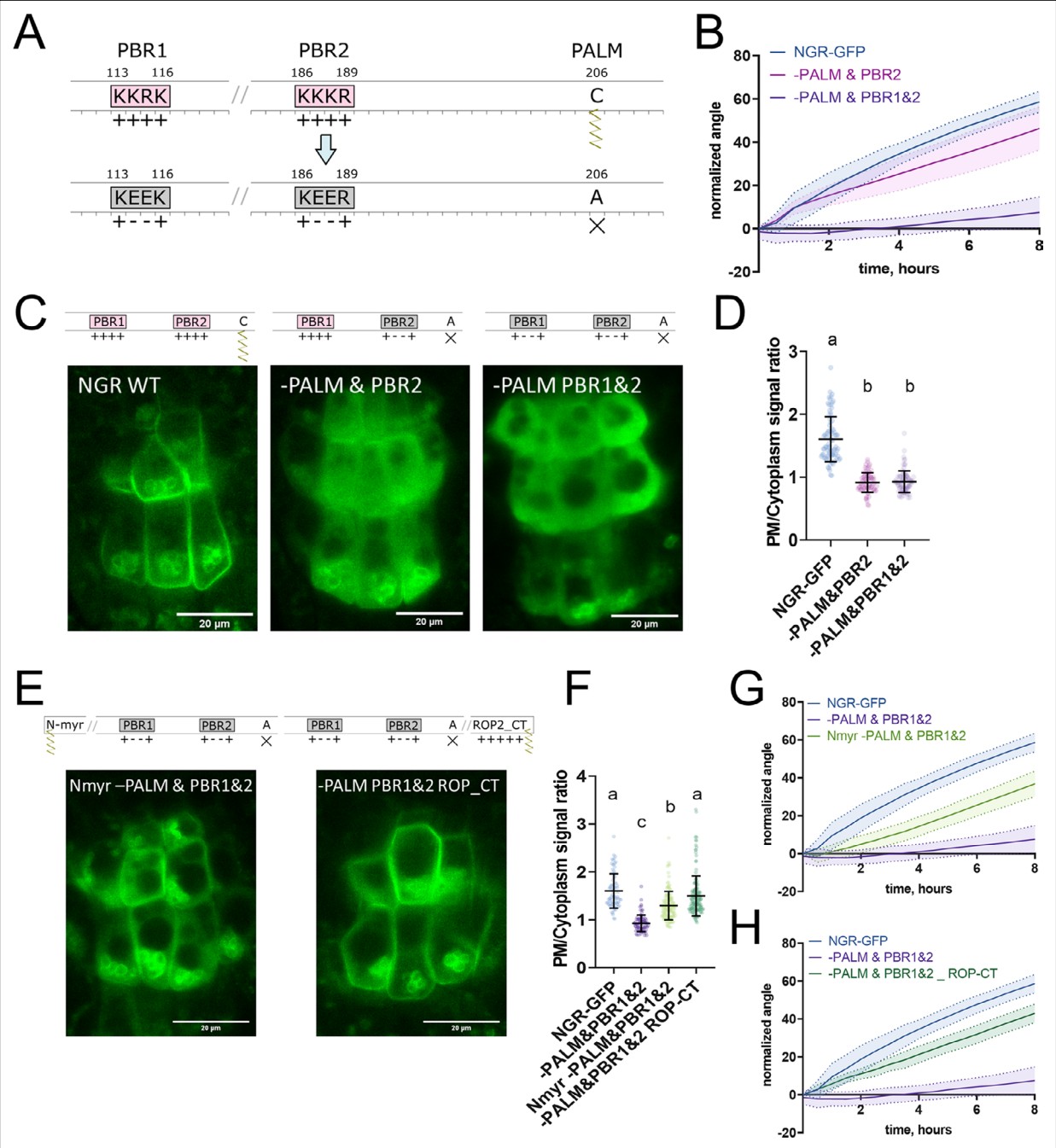

**Figure 3.** Polybasic regions and a palmitoylation site mediate NGR1 membrane localization. (**A**) Scheme of mutagenesis. K114E&R115E (-PBR1), K187E&K188E (-PBR2), and C206A (-PALM) were introduced to disrupt NGR1 polybasic regions and palmitoylation site. Different combinations of mutations were tested. (**B**) Main root angle restoration upon reorientation in the vertical scanner setup of the *ngr1/2/3* triple mutant complemented with *NGR1p::NGR1^-PALM&PBR2^-GFP* and *NGR1p::NGR1^-PALM&PBR1&2^-GFP* in comparison to *NGR1p::NGR1^WT^-GFP*, Col-0 and *ngr1/2/3* triple mutant. (**C**) Representative confocal microscopy images of the root columella cells of *NGR1p::NGR1-GFP*, *NGR1p::NGR1^-PALM&PBR2^-GFP*, and *NGR1p::NGR1^-PALM&PBR1&2^-GFP*. (**D**) PM/cytoplasm signal ratio for NGR1p::NGR1-GFP, NGR1p::NGR1^-PALM&PBR2^-GFP, and NGR1p::NGR1^-PALM&PBR1&2^-GFP variants. Mutation in the polybasic region and palmitoylation site leads to the dissociation of NGR1 protein from the PM. For statistical analysis one-way ANOVA test was used, 'a' is significantly different from 'b' with a p-value <0.0001. The difference between the PM/Cytoplasm signal ratio of NGR1p::NGR1^-PALM&PBR2^-GFP and NGR1p::NGR1^-PALM&PBR1&2^-GFP is not statistically significant. (**E**) Representative pictures of the root columella cells of *NGR1^Nmyr -PALM&PBR1&2^-GFP* and *NGR1^-PALM&PBR1&2 ROP-CT^-GFP*. Membrane localization of NGR1^-PALM&PBR1&2^-GFP is restored upon the addition of LZY1 N-terminal myristoylation site (left) or C-terminal region of ROP2 bearing polybasic region and geranyl-geranylation site (right). (**F**) PM/cytoplasm ratio quantified for the versions with another acylation site. Different letters designate the statistically significant difference with p<0.0001 in a one-way ANOVA test. For each genotype >35 cells were quantified. (**G**) Rescued main root angle restoration upon reorientation in the vertical scanner setup of the *NGR1p::NGR1^Nmyr -PALM&PBR1&2^-GFP* in

*Figure 3 continued on next page*

*Figure 3 continued*

*ngr1/2/3* line, after the addition of LZY1 N-myristoylation site to the NGR1 version with mutated polybasic regions and palmitoylation site. (**H**) Rescued gravitropic response of the *NGR1p::NGR1^{-PALM&PBR1&2 ROP-CT}-GFP* in *ngr1/2/3*, with the addition of the C-terminus of the ROP2 after the GFP tag. For all the mutant variants more than five separate transgenic lines were observed with similar results. 8–15 roots were quantified for each gravitropic bending experiment. Shaded regions depict SD.

The online version of this article includes the following source data and figure supplement(s) for figure 3:

**Source data 1.** Data used for generating the graphs in the figure.

**Figure supplement 1.** Mutations only in PALM site or PBRs are insufficient to disrupt NGR1 function and localization.

**Figure supplement 1—source data 1.** Data used for generating the graphs in the figure.

**Figure supplement 2.** Recovery of NGR1^{mut}-GFP function and localization upon addition of another acylation site.

**Figure supplement 2—source data 1.** Data used for generating the graphs in the figure.

*NGR1^{WT}-GFP* version, and *NGR1^{-PBR2}* and *NGR1^{-PBR1&2}* versions demonstrated a slight delay in the main root angle restoration after reorientation in the vertical scanner setup (*Figure 3—figure supplement 1B*). Then we performed confocal imaging of the root columella cells to check the localization of the NGR1^{-PALM}-GFP, NGR1^{-PBR2}-GFP, and NGR1^{-PBR1&2}-GFP proteins (*Figure 3—figure supplement 1C*). To compare the different mutant versions, we quantified the PM/cytoplasm signal ratio. The PM/cytoplasm ratio was slightly decreased for NGR1^{-PALM}-GFP, NGR1^{-PBR2}-GFP, and NGR1^{-PBR1&2}-GFP mutants compared to NGR1^{WT}-GFP (*Figure 3—figure supplement 1D*).

As the mutations only in the palmitoylation site or only in the PBRs of the NGR1 did not lead to the significant disruption of the plant gravitropic response and the dissociation of the protein from the membrane, we combined these mutations to check their collective effect. We generated NGR1^{-PALM&PBR2}-GFP with a mutation in the PALM site and the PBR2, and NGR1^{-PALM&PBR1&2} with a mutation in both PBR1 and PBR2 and PALM site. *NGR1p::NGR1^{-PALM&PBR2}-GFP* and *NGR1p::NGR1^{-PALM&PBR1&2}-GFP* could not fully complement *ngr1/2/3* triple mutant. The main root of the 5 d.o. seedlings demonstrated inclination from the vertical angle (*Figure 3—figure supplement 1E*). In contrast to *NGR1-GFP* complemented *ngr1/2/3* plants, complementation with *NGR1^{-PALM&PBR2}* displayed partially compromised restoration of the main root angle upon reorientation, while the *NGR1^{-PALM&PBR1&2}* root bending was completely disrupted (*Figure 3B*). With confocal microscopy we observed the dissociation of the NGR1^{-PALM&PBR2} and NGR1^{-PALM&PBR1&2} from the PM (*Figure 3C*). PM/cytoplasm ratio was significantly decreased for both NGR1^{-PALM&PBR2} and NGR1^{-PALM&PBR1&2} compared to non-mutated NGR1-GFP (*Figure 3D*). Taken together, the combination of mutations in PBRs and palmitoylation site resulted in the decrease of NGR1-GFP PM binding and loss of the protein functionality.

If the selected PBRs and palmitoylation site represent solely the mean of the PM anchoring, then bringing the protein back to the PM by different targeting signal should restore its functionality. To test this hypothesis, we introduced another membrane-association site to the mutant versions of NGR1-GFP. It was shown that the N-terminus of LZY1 mediates its membrane localization potentially via a predicted N-myristoylation site (*Yoshihara and Spalding, 2020*). This site is not conserved in the NGR1. Remarkably, LZY1 does not possess the palmitoylated Cys residue conserved in NGR1 (*Figure 3—figure supplement 2A*). We changed the N-terminus of all the NGR1 mutant versions carrying the mutation C^{206}A to introduce the LZY1 N-terminal myristoylation site. The obtained *NGR1^{Nmyr -PALM}-GFP*, *NGR1^{Nmyr -PALM&PBR2}-GFP*, *NGR1^{Nmyr -PALM&PBR1&2}-GFP* versions under *NGR1* native promoter were transformed to the *ngr1/2/3*.

As an alternative, we used a well-characterized membrane-association sequence from the C-terminus of ROP2 (ROP-CT) carrying a PBR and geranyl-geranylation site (*Figure 3—figure supplement 2B*; *Lin et al., 1996*; *Yalovsky et al., 2008*). Similarly, as for the N-myristoylation site, we added the ROP-CT on the C-terminus after the GFP tag to all the variants with the mutated PALM site. The resulting *NGR1p::NGR1^{-PALM ROP-CT}-GFP, NGR1p::NGR1^{-PALM&PBR2 ROP-CT}-GFP, NGR1p::NGR1^{-PALM&PBR1&2 ROP-CT}-GFP* were transformed into *ngr1/2/3* background. PM localization and gravitropic response of the versions with another acylation site were restored gradually (*Figure 3—figure supplement 2C–H*).

We examined the localization of the GFP signal in the columella of both restored lines (*NGR1p::NGR1^{Nmyr -PALM&PBR1&2}-GFP* and *NGR1p::NGR1^{-PALM&PBR1&2 ROP-CT}-GFP*). NGR1^{Nmyr -PALM&PBR1&2} membrane association was partially restored, and NGR1^{-PALM&PBR1&2 ROP-CT} membrane association was fully restored

(*Figure 3E and F*). In accordance with PM localization, the gravitropic response of those plants was also restored (*Figure 3G and H*).

Thus NGR1 PM localization is mediated synergistically by PBRs and palmitoylation site, and the NGR1 PM association is crucial for NGR1 functioning in root gravitropic response.

## D6PK relocalizes in columella following gravistimulation in NGR1-dependent manner

NGR1 protein relocalization occurs simultaneously with statolith sedimentation and is rapidly followed by PIN-dependent asymmetric auxin accumulation at the lower root side. These events, however, precede a visible PIN3 polarization towards the bottom columella cell sides (*Figure 1D–G*). Alternatively, we hypothesized that asymmetric PIN activation at this position may precede the PIN polarization there and mediate the early auxin asymmetry. To test this, we analyzed the localization of D6PK, a well-established PIN3 activating kinase (*Willige et al., 2013*). We monitored D6PK under its native promoter with N-terminal YFP and mCHERRY (mCH) tags. This revealed a highly dynamic polarized D6PK localization following gravistimulation. Following 180° rotation, YFP-D6PK and mCH-D6PK displayed a very clear translocation to the new bottom side of the columella cells (*Figure 4A–C*). To resolve whether this polarization depends on NGR proteins, we transformed mCHERRY-D6PK (mCH-D6PK) to both *ngr1/2/3* triple mutants and Col-0 wild type plants. Notably, following gravistimulation, D6PK polarization was absent in *ngr1/2/3* triple mutants (*Figure 4C–D*, *Video 2*). Surprisingly, other cell types such as meristematic cells or epidermal cells still displayed undisturbed basal (WT-like) D6PK polarity in *ngr1/2/3* triple mutant suggesting a role of NGRs in D6PK targeting specifically during gravity response (*Figure 4C*). To compare D6PK translocation with other events during gravity response, we generated a movie, where polarization of D6PK is compared with the calcium transients, PIN3 mobility, and DII Venus signal (*Video 3*).

Taken together, we identified D6PK, a PIN activating kinase to be translocated towards the bottom columella cell sides in an NGR-dependent manner. This provides a plausible mechanism for rapidly redirecting auxin fluxes prior to PIN relocation during root gravitropic response.

## Discussion

In this study, we provide several new crucial insights into the mechanism through which plant roots adapt their growth in response to gravity: (i) we mapped the early events after gravistimulation with unprecedented time resolution challenging the current gravitropic models; (ii) we observed real-time relocalization of the key gravitropic regulator, NGR1 on and alongside the descending statoliths and (iii) mapped the conditions for NGR1 crucial association with the PM; and, finally, (iv) we detected rapid, NGR1-dependent relocation of D6PK auxin transport activator. This altogether suggests a new mechanism for rapid redirection of auxin fluxes following gravistimulation.

### Early events during root gravitropic response

The temporal correlation between NGR1 relocation and auxin asymmetry dynamics was tested by monitoring auxin-dependent calcium transients. Interestingly, while an auxin-induced calcium wave can be observed following statolith descent to the bottom side of the gravity-sensing cells, preceding this, a decline in calcium transients occurs at the upper root side. These observations suggest that the first signaling event in root gravitropic bending may not be statolith contact with the cell bottom, but rather the departure of statoliths from the upper PM, closely followed by NGR1 and D6PK translocation to the bottom cell sides. Only significantly later, similar polarization of PIN auxin exporters can be confidently observed. This may be reflected by actual onset of the root bending, as there is growing evidence, that the actual root bending begins much sooner than previously reported, in fact, some degree of bending was observed already within the first 5 min following gravistimulation (*Dubey et al., 2023*; *Serre et al., 2023*).

### Insights into NGR membrane association and gravity-induced polarization

NGR-GFP relocation to the new bottom membrane happens rapidly and is not affected when ARF GEF-mediated endomembrane trafficking is inhibited. Although the precise mechanism of NGR1

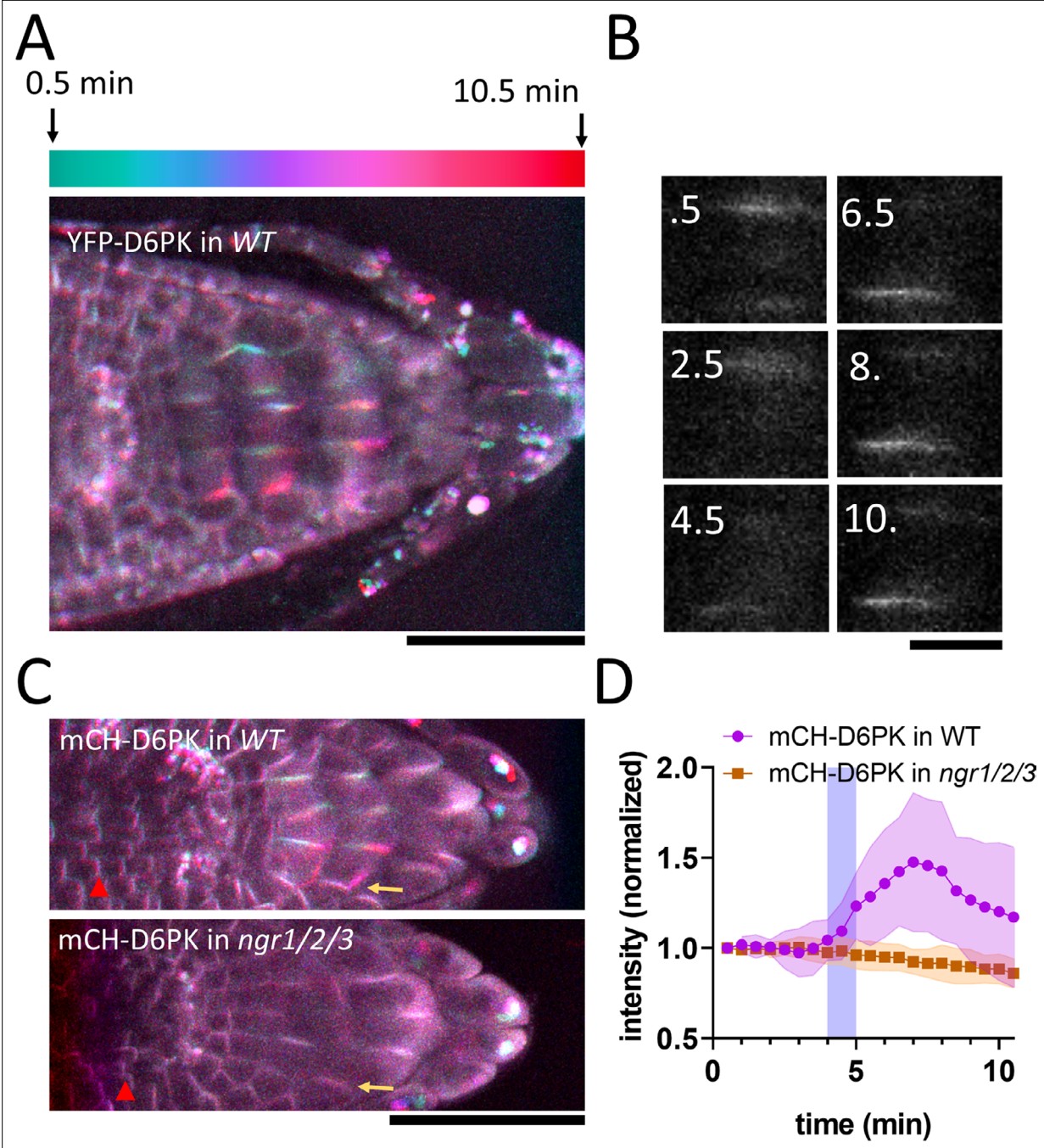

**Figure 4.** NGR-dependent D6 protein kinase translocation. (**A**) Temporal color-coded image of YFP-D6PK translocation induced by 180° rotation. (**B**) Single gravistimulated (180°) columella cell undergoing YFP-D6PK translocation. White numbers indicate time after rotation (min). (**C**) mCH-D6PK 10 min after gravistimulation in WT and *ngr1/2/3* triple mutant, same temporal color code as in A. Yellow arrow depicts the new bottom membrane. Red arrowhead depicts meristematic cells, which displayed basal D6PK polarity in both genotypes. (**D**) Quantification of D6PK signal at the columella cell bottom PM upon 180° rotation. Blue strip depicts approximate time of statolith contact with the bottom PM. Bars = 50 µm (**A,C**) 10 µm (**B**). Ten cells out of three roots were quantified. Experiment was replicated three times with similar results. Shaded regions depict SD.

The online version of this article includes the following source data for figure 4:

**Source data 1.** Data used for generating the graphs in the figure.

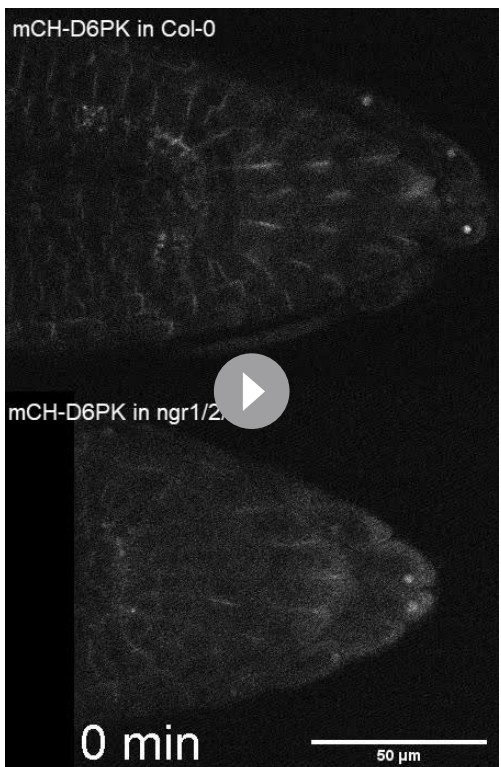

**Video 2.** mCHERRY-D6PK in WT and *ngr1/2/3* mutant upon gravitropic stimulus.
https://elifesciences.org/articles/91523/figures#video2

dynamic relocation following gravitropic stimulus still remains elusive, this study brings some new insights. The NGR1 membrane anchoring mechanism is dependent on cysteine acylation and adjacent polybasic stretches, which interact with charged lipids at the membrane surface. Such a mechanism is typical for a plethora of proteins (*Hemsley and Grierson, 2008*). Since we could restore the protein functionality by adding two different membrane anchors on the NGR1 N and C terminus, we conclude that the mechanism of membrane targeting is rather universal. Point mutated NGR1 proteins which lost their membrane binding ability still retained the statolith binding, implying an independent targeting mechanism. Therefore, we can conclude that statolith-bound NGR1 is not sufficient for the NGR1 function in gravitropism. On the other hand, NGR amyloplast association is apparently an essential aspect of the NGR mobility. While writing this manuscript, two concurrent articles have addressed the mechanism of NGR starch grain association and its translocation to the plasma membrane. These studies reveal that the mobility of the NGR (LAZY4) protein depends on the N-terminal transit peptides. The association of LAZY4 with the translocons in the outer plastid membrane is modulated by mitogen-activated protein kinases, facilitating its dynamic transfer to the plasma membrane (*Nishimura et al., 2023*; *Chen et al., 2023*) with a specific lipid composition.

Our pharmacological experiments indicate that both sterols and charged lipids play a role in the NGR membrane binding. This resembles other highly polarized structures such as root hairs, where sterols and phosphatidylinositol 4,5-bisphosphate are vital for the polarity (*Champeyroux et al., 2020*; *Ovecka et al., 2010*). Furthermore, PIN2 polarity was defective in sterol synthesis mutants, implying the role of sterols in planar polarity as well (*Men et al., 2008*). Whether there is mobile sterol-rich domain moving alongside the statolith or the membrane composition is uniform remains to be elucidated.

## Gravity-induced D6PK kinase polarization as mechanism for rapid auxin fluxes redirection

PIN auxin transporters were shown to polarize following gravistimulation to the bottom of gravity-sensing cells (*Friml et al., 2002*; *Furutani et al., 2020*; *Kleine-Vehn et al., 2010*) providing a mechanism for redirection of auxin fluxes downwards. Although PIN polarization has been shown to be reliant on NGR proteins (*Ge and Chen, 2019*), the timing of early gravitropic events resolved here suggests that it occurs too late to account for the initial auxin asymmetry. Either,

**Video 3.** Summary of the timing of gravitropic bending for YFP-D6PK, PIN3, calcium transients, and DII-Venus responses during a series of 180° flips.
https://elifesciences.org/articles/91523/figures#video3

we are unable to detect early, less-pronounced PIN asymmetry or there is an additional mechanism preceding the auxin flux redirection via PIN relocation.

Our discovery of similar gravity-induced polarization of D6PK AGCVIII kinases, which is contingent on NGR activity, bridges this critical knowledge gap and suggests that, prior to relocation of PIN transporters, AGCVIII-mediated PIN activation at the lower cell sides rapidly redirects auxin fluxes following gravity perception. Notably, PIN3 phosphorylation is important for gravity response (*Grones et al., 2018*) and higher order D6PK mutants display mild gravitropic bending phenotypes (*Zourelidou et al., 2009*), which are, however, weaker than *ngr1/2/3* mutants. This observation might be attributed to the multiplicity of the AGC kinase family and/or an additional mechanism governing the differential accumulation of PIN proteins, following D6PK polarization, thus contributing to the regulation of auxin flux.

Similarly to NGRs, D6PK membrane localization depends on PBRs and sterol-rich environments (*Barbosa et al., 2016*; *Stanislas et al., 2015*). However, in contrast to NGR proteins, D6PK localization is dependent on BFA-sensitive trafficking regulator GNOM (*Barbosa et al., 2016*; *Barbosa et al., 2014*) and thus is likely delivered to the PM by vesicle trafficking. RLD proteins, implicated in the regulation of the GNOM-dependent PIN trafficking pathway, are directly recruited by NGR homolog NGR3 (LZY3; *Furutani et al., 2020*). Notably, RLD proteins contain the BRX domain, which is responsible for the NGR interaction (*Furutani et al., 2020*). In the different cellular context, BRX protein was shown to be an inhibitor of AGC kinase (PAX) induced PIN3 activation and an interactor of multiple D6PK proteins (*Marhava et al., 2018*). This implies that BRX containing proteins may act as negative feedback on PIN phosphorylation after gravistimulation. Indeed, our observations indicate that the D6PK signal at the new bottom membrane reaches its maximum approximately 7–8 min after gravistimulation and subsequently diminishes, hinting at the existence of a negative feedback mechanism. The involvement of RLDs in this process remains to be investigated.

Taken together, an updated model of early events in root gravity-sensing cells emerges: (i) Statoliths sediment and bring associated NGR proteins to the lower PM; by a mechanism requiring electrostatic interaction-based association of NGR with the PM; (ii) dependent on these events, the D6PK translocates almost concomitantly also to the bottom cell sides, likely by GNOM-dependent trafficking mechanism; (iii) D6PK activates PINs at the lower side, auxin transport get redirected downwards reflected by a decrease of auxin response at the upper side followed by auxin response increase at the lower root side; (iv) D6PK asymmetry gradually diminishes, possibly by action of BRX domain-containing proteins and the whole PIN-dependent auxin transport machinery gradually relocates to the lower cell side for more sustained redirection of auxin fluxes. There are still many mechanistic links to be confirmed and clarified in this model but it is consistent with all observations on the sequence of events following gravistimulation and with all additional data from other developmental contexts.

## Materials and methods

### Plant cultivation

All *Arabidopsis* mutants and transgenic lines which were used in this project are in the Columbia-0 (Col-0) background. *NGR1p::NGR1-GFP*, *35Sp::NGR1-GFP*, and *D6PKp::mCH-D6PK* were generated as part of this research project. *D6PKp::YFP-D6PK* has been described previously (*Zourelidou et al., 2009*). Overnight seed sterilization was executed with chlorine gas. Seeds were sown on solid agarose media – half-strength Murashige and Skoog medium supplemented with 1% sucrose (AM+) and 0.8% phyto agar (pH = 5.9) – stratified at 4 °C for 2 days and subsequently grown vertically at 21 °C with 16 hr light/8 hr dark cycles.

### Cloning strategy

*NGR1* promoter was amplified from genomic DNA using the primer pairs NGR1p-B4-FP and NGR1p-B1r-RP, and was cloned into pDONR-P4P1r. *NGR1-GFP* fusion fragment was obtained by overlapping PCR using the prime pairs NGR1g-B1-FP and NGR1g-L-RP to amplify the *NGR1* protein-coding region with genomic DNA as the template and the primer pairs eGFP-L-FP and eGFP-B2-RP to amplify the *eGFP* coding sequence, with a PKPA protein linker inserted between NGR1 and GFP, and the resulting fragment was cloned into pDONR221. The resulting pENTRY clones were recombined into

pH7m24GW to get the final expression construct *NGR1p::NGR1-GFP*. The pENTR clone NGR1-GFP in pDONR221 was recombined with pH2GW7 to get the construct *35Sp::NGR1-GFP*.

To obtain desired point mutations in PBRs and acylation sites, Mutagenesis of *NGR1p::NGR1-GFP* was performed using Gibson assembly (NEB HiFi DNA assembly E2621L), overlapping compatible cohesive ends were modified. For ROP C-terminus, an oligo-bridge Gibson assembly was performed with an oligonucleotide containing the entire ROP2 hypervariable region. Plants were transformed with floral dip method (*Zhang et al., 2006*).

*NGR1* transformants were selected on plates containing 30 μg/ml hygromycin. *NGR1p::NGR1-GFP* plants with genotyped with primers gNGRmut-NGR1prom-fwd and gNGRmut-T35S-rev, and the middle part of the NGR1 was sequenced to confirm the presence of the mutations.

*D6PKp::mCH-D6PK* was generated using the Greengate cloning strategy. Greengate entry blocks were generated with the use of the Gibson assembly (pGGA-D6PK promotor) and restriction-ligation assembly (pGGC-D6PK). Two KB fragment upstream of D6PK start codon was used and the BsaI site was removed using point mutated compatible cohesive ends for the Gibson assembly between fragments A and B. Other building blocks used were: pGGB-mCHERRY-linker (pGGB001), pGGD-Dummy (pGGD002), pGGE UBQ10 terminator (pGGE009), pGGF-d-AlanineR (pGGF003), destination vector (pGGZ001 modified to contain bacterial kanamycin resistance; *Lampropoulos et al., 2013*), plasmids were obtained from Addgene or were kindly provided by Dr. Andrea Bleckmann. *pD6PK::mCH-D6PK* transformants bearing DAO-selection cassette were selected on the 3 mM D-Alanine.

## Microscopy

For imaging, vertically mounted Zeiss LSM800 was used with air objectives Plan Apochromat 20 x/0.8 M27 and Plan Apochromat 10 x/0.45 M27 (*Figure 1G*, *Video 3*). Air objective Plan Apochromat 20 x/0.8 M27 and water objective Plan-Apochromat 40 x/1.2 M27 were used in *Figure 2*, *Figure 1—figure supplement 1*, *Figure 2—figure supplement 1*, *Figure 3*, *Figure 3—figure supplements 1 and 2*, *Figure 4*, *Video 2* and *Video 3*. Plan apochromat 60x1.4 Oil objective was used for the airyscan imaging (*Figure 1—figure supplement 1C*). For fluorescence lifetime imaging (*Figure 1—figure supplement 1B*), Leica Stellaris 5 with Plan-Apochromat 40 x/1.2 M27 was used. GCaMP signal was imaged using GFP emission (500–540 nm) using 405 nm and 488 nm excitation lasers. Signal obtained by 405 nm excitation was used as a ratiometric reference. For the gravitropic experiments, brightfield images were always taken together with the fluorescent probes to determine the statolith contact with basal PM. This was determined as a time at which statolith movement relative to the cell surface stopped.

## Plant treatments

The following inhibitors were dissolved in DMSO: Brefeldin A (Sigma B7651); Cycloheximide (Sigma C1988); PAO (Sigma P3075); FEN (Sigma 36772); LOV (Sigma PHR1285); NPA (DUCHEFA N0926). BFA treatment was performed as described previously (*Geldner et al., 2001*). In brief, seedlings were submerged in liquid AM + medium and pre-treated with 50 μM CHX for 30 minutes. Subsequently, they were incubated with 50 μM BFA and 50 μM CHX for 75 minutes. Then, 2 μM FM4-64 (Invitrogen T13320) was added and co-incubated with 50 μM BFA and 50 μM CHX for 15 minutes. Treatment with PAO was executed as described previously *Barbosa et al., 2016*; seedlings were incubated in 30 μM PAO-containing liquid AM + medium for 20 min, followed by the addition of 2 μM FM4-64 and another 10-min incubation. For quantification purposes, 5-day-old seedlings grown on AM + plates with 1 μM LOV and 250 μg/ml FEN were utilized (*Stanislas et al., 2015*). Representative images were captured of 5-day-old seedlings that were initially grown on AM + and subsequently transferred to 1 μM LOV and 50 μg/ml FEN for a duration of 2 hr. To treat with DTT, seedlings were incubated in liquid AM + medium supplemented with 50 mM DTT (Sigma; dissolved in H2O) for 3 hr, followed by a 10-min incubation with 2 μM FM4-64 (*Sabol et al., 2017*). Unless otherwise stated, treated seedlings were mounted in liquid AM + medium for live imaging. As a control for used inhibitors, solvent control treatments were performed in parallel.

## Data analysis

To quantify signals in the individual cells, time series were stabilized by Correct 3D drift Fiji plugin. All image data was obtained in 16-bit format. ROI were selected and measured using Microsoft Excel and

GraphPad Prism. Data was normalized so that the first value = 1 (*Figures 1A, D, F and 4D*) or by using reference channel (*Figures 1E and 2B*).

In order to analyze subcellular colocalization we utilized the tool JACoP (Just Another Co-localization Plugin), that allows for a correlation of pixels of dual-channel images based on Pearson's coefficient (*Bolte and Cordelières, 2006*).

Gravitropic bending experiments were performed in the vertically oriented scanner setup. 4 d.o. seedlings were transferred to a new solid agarose media plate and aligned so that the roots were straightened. Images obtained with 30 min intervals were registered with the Fiji StackReg plugin (*Schindelin et al., 2012*). Roots were manually tracked and angle increments were calculated using Microsoft Excel.

For the PM/cytoplasm ratio quantification, the background of the confocal images was subtracted; with the multi-point selection tool 15–50 points on the PM, and 15–50 points in the cytoplasm were selected, and the intensity of the GFP signal was measured.

The location of the PM membrane was confirmed with the FM4-64 staining. The PM region was selected on the bottom side of the cells, as NGR1 is localized polarly. The cytoplasmic region was selected close to the selected membrane region and did not include the region with amyloplasts and the vacuole. For individual cells the averaged PM signal was divided by the averaged cytoplasmic signal. The PM/cytoplasm ratio for the lines is quantified as the average of the PM/cytoplasm ratios for individual cells of this line.

If not stated otherwise, all experiments were performed in triplicate. Three or more independent transgenic lines were always used with similar results. All error bars and shaded areas around average represent SD.

## List of primers used in the study

| Primer name | Primer sequence |
| --- | --- |
| **Transgenic line preparation** | |
| NGR1p-B4-FP | GGGGACAACTTTGTATAGAAAAGTTGGAAGAGGAGAAGGTGGGAGAGC |
| NGR1p-B1r-RP | GGGGACTGCTTTTTTGTACAAACTTGTGTTTCTTTTTTTCTGACAATTGACTG |
| NGR1p-S-F1 | ACTAAACAACCCCTTTTGAAACC |
| NGR1p-S-F2 | CAAGATTGAAAAACTATTTGCCCT |
| NGR1g-B1-FP | GGGGACAAGTTTGTACAAAAAAGCAGGCTCCATGAAGTTCTTCGGGTGGATG |
| NGR1g-L-RP | CACCATAGCAGGCTTAGGTATCTCGAGAACTATGACTGTAATCA |
| eGFP-L-FP | GAGATACCTAAGCCTGCTATGGTGAGCAAGGGCGAGGAG |
| eGFP-B2-RP | GGGGACCACTTTGTACAAGAAAGCTGGGTGCTACTTGTACAGCTCGTCCATGCC |
| pGGC_D6PK _for | AGAAGTGAAGCTTGGTCTCAGGCTCCATGATGGCTTCAAAAACTCC |
| pGGC_D6PK_rev | AGGGCGAGAATTCGGTCTCACTGATCAGAAGAAATCAAACTCAAGATA |
| pGGA_D6PKpromA_for | GTGAAGCTTGGTCTCAACCTCTGTTGAACCATTTCTAAAAAAAC |
| pGGA_D6PKpromA_rev | GAGAGAGAGTCCCAATAAATCGTTACCTG |
| pGGA_D6PKpromB_for | ATTTATTGGGACTCTCTCTCTCTCTCTC |
| pGGA_D6PKpromB_rev | CGAGAATTCGGTCTCATGTTTAACACAGAGCAATCTTAAAC |
| pGGA_backbone_for | AACATGAGACCGAATTCTC |
| pGGA_backbone_rev | AGGTTGAGACCAAGCTTC |
| **Genotyping** | |
| ngr1-F199G07-GT-FP | GCGCAGACAAAAATCTTCTTG |
| ngr1-F199G07-GT-RP | TTGGTGGACTCGTTTGCTTAC |
| FLAG-LB4 | CGTGTGCCAGGTGCCCACGGAATAGT |
| ngr2-SAIL723H11-GT-FP | TTTGGTTTTATGGACCCAACC |

*Continued on next page*

*Continued*

| Primer name | Primer sequence |
|---|---|
| ngr2-SAIL723H11-GT-RP | AAGAGCTTTCTTCCTCCGATG |
| SAIL-LB3 | TAGCATCTGAATTTCATAACCAATCTCGATACAC |
| ngr3-GK479C08-GT-FP | GCAAACAGATTTTCTTCACCAC |
| ngr3-GK479C08-GT-RP | GCACAAGTGGCTTCAAAACTC |
| GABI-O8409 | ATATTGACCATCATACTCATTGC |
| gNGRmut-NGR1prom-fwd | GTAGTCAAAGTTTGGAACTTGAACACC |
| gNGRmut-T35S-rev | CTGGGAACTACTCACACATTATTCTGG |
| Mutagenesis | |
| mutatePBR2-fwd | TAAGAATAACAAGGAAGAAAGAGACATAAGCAAGAACTCTG |
| mutatePBR2-rev | TCTTTCTTCCTTGTTATTCTTACTCTCTATTGATATCTC |
| PBR1&2-fragment1-bb-fwd | TAAGAATAACAAGGAAGAAAGAGACATAAGCAAGAACTCTG |
| PBR1&2-fragment1-bb-rev | TCACATCAGACTTTTCTTCCTTAGTTCTTGACAAGAGCTTC |
| PBR1&2-fragment2-insert-fwd | AAGGAAGAAAGTCTGATGTGAATAGAGAGC |
| PBR1&2-fragment2-insert-rev | TCTTTCTTCCTTGTTATTCTTACTCTCTATTGATATCTC |
| mutatePALM_C206A-fwd | CAAGAAGATTTTTGTCGCTGCAGATGGG |
| mutatePALM_C206A-rev | CAGCGACAAAAATCTTCTTGAAAAGATATGAGACAGAG |
| add-Nmyr-fwd | CCACAAGTTCAGGGGGGATCATAACAGAACAAGCACTTCC |
| add-Nmyr-rev | CCTGAACTTGTGGTGCATCCACCCCCAGAACTTCATGGAGCC |
| ROP2-CT-oligo-bridge | GCTATAAAAGTGGTGCTTCAGCCACCAAAGCAGAAGAAGA AGAAAAAGAATAAGAACCGTTGCGCGTTCTTGTGA |
| NGR-ROP2_CT-after-GFP-fwd | ACCGTTGCGCGTTCTTGTGACACCCAGCTTTCTTGTAC |
| NGR-ROP2_CT-after-GFP-rev | TGAAGCACCACTTTTATAGCCTTGTACAGCTCGTCCATG |

All the transgenic lines generated in this study can be accessed by contacting the corresponding author.

## Acknowledgements

The research leading to these results has received funding from the European Research Council (ERC) under the European Union's Horizon 2020 research and innovation programme grant agreement No 742985 and Austrian Science Fund (FWF): I3630-775 B25 to J.F. This research was also supported by the Lab Support Facility (LSF) and the Imaging and Optics Facility (IOF) of IST Austria, namely Tereza Bělinová for her help with the imaging. JS was supported by FemTECH fellowship.

## Additional information

### Funding

| Funder | Grant reference number | Author |
|---|---|---|
| Horizon 2020 Framework Programme | No 742985 | Jiří Friml |
| FEMtech Internships for Female Students (Federal Ministry Republic of Austria for Climate Action, Environment, Energy, Mobility, Innovation and Technology) | | Julia Schmid |

| Funder | Grant reference number | Author |
|---|---|---|
| Austrian Science Fund | I3630-775 B25 | Jiří Friml |
| FemTECH fellowship | | Julia Schmid |

The funders had no role in study design, data collection and interpretation, or the decision to submit the work for publication.

## Author contributions

Ivan Kulich, Conceptualization, Data curation, Formal analysis, Supervision, Investigation, Methodology, Writing – original draft; Julia Schmid, Formal analysis, Investigation, Methodology, Writing – original draft, Writing - review and editing; Anastasia Teplova, Investigation, Methodology, Writing – original draft, Writing - review and editing; Linlin Qi, Conceptualization, Resources, Investigation, Methodology; Jiří Friml, Resources, Supervision, Funding acquisition, Writing – original draft, Project administration

## Author ORCIDs

Ivan Kulich ⓘ http://orcid.org/0000-0002-0458-6470
Anastasia Teplova ⓘ http://orcid.org/0009-0009-4102-3798
Jiří Friml ⓘ http://orcid.org/0000-0002-8302-7596

Reviewer #1 (Public Review): https://doi.org/10.7554/eLife.91523.3.sa1
Reviewer #2 (Public Review): https://doi.org/10.7554/eLife.91523.3.sa2
Author Response https://doi.org/10.7554/eLife.91523.3.sa3

# Additional files

## Supplementary files

- MDAR checklist

## Data availability

All data generated in this study are included as a manuscript supporting files.

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
