## [Editor Report · eLife assessment]

This **fundamental** study addresses the earliest events that enable plant roots to reorient growth in response to gravity. **Compelling** molecular and cell biological data establish that plasma membrane localization of the LAZY or NEGATIVE GRAVITROPIC RESPONSE OF ROOTS (NGR) protein family is required for rapid and polar redirection of D6 protein kinase, an activator of the PIN3 auxin transporter. This work complements and extends recent publications on the NGR family in gravity sensing (PMID: 37741279 and PMID: 37561884). Collectively these papers advance our understanding of rapid plant gravity sensing and response.

---

## [Referee Report · Reviewer #1 (Public Review)]

Summary:

Plant roots grow following the gravity vector. Changes in the direction of gravity can be sensed in the root tip by specialized cells that hold starch granules. These starch granules act as levels. Movement and settling of the granules at the bottom of these specialized cells initiates an imbalanced distribution of auxin, a key hormone for plant development. Consequently, this leads to a reorientation of root growth towards the newly established gravity vector. This work provides new insights into granules' relocalization, the proteins associated with them, and the molecular processes triggered downstream.

Comments on revised submission:

In the previous review round, the reviewers noted that the authors had missed an opportunity to discuss the results presented in two recently published articles closely related to the topic of their manuscript. The authors have now referenced these articles in the current version of the manuscript, but the discussion remains rather brief. It would have been beneficial to summarize, identify, and highlight the similarities among these studies in a more comprehensive manner.

In Figure 1, it would have been more informative if the authors had provided specific information concerning the key time-points described in the graphs to visually illustrate the dynamics of PIN3 localization, intracellular calcium transients, and auxin reporter DII Venus. Including these images would have perfectly complemented panels E, F, and G.

The authors expressed concerns about overcrowding the figure. If the aesthetics of the figure were their primary concern, they could have included essential image frames for the data represented in the graphs in a supplementary figure. Alternatively, a detailed description of supplementary movie 3, highlighting the specific frames quantified in the graphs (Figure 1), could have sufficed.

---

## [Referee Report · Reviewer #2 (Public Review)]

Summary:

This manuscript addresses what rapid molecular events underly the earliest responses after gravity-sensing via the sedimentation of starch-enriched amyloplasts in columella cells of the plant root cap. The LAZY or NEGATIVE GRAVITROPIC RESPONSE OF ROOTS (NGR) protein family is involved in this process and localizes to both the amyloplast and to the plasma membrane (PM) of columella cells.

This manuscript complements and extends a very recent study, (Nishimura et al., Science, 2023, August 10, 2023) that reported that the LZY3 and LZY4 proteins translocate from amyloplasts to the PM and that this translocation is likely necessary for the root gravitropic response. Kulich and colleagues describe the role of the LZY2 protein, also called NGR1, during this process, imaging its fast relocation and addressing additional novel points such as molecular mechanisms underlying NGR1 plasma membrane association as well as revealing the requirement of NGR1/LZY2, 3,4 for the polar localization of the AGCVIII D6 protein kinase at the PM of columella cells, in which NGR1/LZY2 acts redundantly with LZY3 and LZY4.

The authors initially monitored relocalization of functional NGR1-GFP in columella cells of the ngr1 ngr2 ngr3 triple mutant after 180 degree reorientation of the roots. Within 10 -15 min NGR1-GFP signal disappeared from the upper PM after reorientation and reappeared at the lower PM of the reoriented cells in close proximity to the sedimented amyloplasts. Reorientation of NGR1-GFP occurred substantially faster than PIN3-GFP reorientation, at about the same time or slightly later than a rise in a calcium sensor (GCaMP3) just preceding a change in D2-Venus auxin sensor alterations. Reorientation of NGR1-GFP proved to be fast and not dependent on a brefeldin A-sensitive ARF GEF-mediated vesicle trafficking, unlike the trafficking of PIN proteins, like PIN3, or the AGCVIII D6 protein kinase. Strikingly, the PM association of NGR1-GFP was highly sensitive to pharmacological interference with sterol composition or concentration and phosphatidylinositol (4)kinase inhibition as well as dithiothreitol (DTT) treatment interfering with thioester bond formation e.g. during S-acylation. Indeed, combined mutation of a palmitoylation site and polybasic regions of NRG1 abolished its PM but not its amyloplast localization and rendered the protein non-functional during the gravitropic response, suggesting NRG1 PM localization is essential for the gravitropic response. Targeting the protein to the PM via an artificially introduced N-terminal myristoylation and a ROP2-derived polybasic region and geranylgeranylation site partially restored its functionality in the gravitropic response.

Strengths:

This timely work should be of broad interest to plant, cell and developmental biologists across the field as gravity sensing and signaling may well be of general interest. The point that NGR1 is rapidly responsive to gravistimulation, polarizes at the PM in the vicinity to amyloplast and that this is required for repolarization of D6 protein kinase, prior to PIN relocation is really compelling. The manuscript is generally well written and accessible to a general readership, except for very minor language errors. The figures are clear and of high quality, the methods are sufficiently explained for reproduction of the experiments.

Comments on revised submission:

The authors have addressed my comments to a large part, however, while they write they have updated the statistical analysis as requested, they only did this for the main figures, but NOT for the supplementary images (except for Fig. S2) and their legends. These issues need fixing in order to correctly describe the data and let the reader know, which distributions actually differed. Some specific examples of concerns are:

In Figs. 3F and D we now know that a one-way ANOVA test was performed and that letters designate the statistically significant difference between distributions with p smaller 0.0001, but we still do not know what "n" in the displayed distributions is e.g. how many PM/cytoplasm ratios were measured i.e. e.g 112? (from 112 cells?). It is said that 8-15 roots were quantified, but the data points in the distributions are not 8-15 .... . They are many more, so, "n" must be the number of cells derived from 8-15 roots but what is "n" in the displayed distributions and is that the same value that was used for the Anova test?

This must be clarified as it has very well been done for Fig. 2 and Fig. S2B, E in the legends and by inserting a lettering for significance differences in the figures.

Similar information is still lacking for Fig. S3D, no number "n" of cells from which the PM/cytoplasm ratios are analyzed is given, no lettering for differences, no p -value. This leaves one to guess which distributions differ from each other.

This also needs to be fixed for Figs. S4 E, F (for G and H one can see the differences where the SDs do not overlap and it is explained what they are derived from).

---

## [Author Response]

The following is the authors’ response to the original reviews.

**Public Reviews:**

**Reviewer #1 (Public Review):**
Summary:The current work by Kulich et al. examines the dynamic relocalization of NGR1 (LAZY2) a member of the LAZY protein family which is key for auxin redistribution during gravitropic responses. After gravistimulation of the triple mutant ngr123 (lazy234), the PIN3 activating kinase D6PK is not polarized in the columella cells.Strengths:The authors show a thorough characterization of NGR1 relocalization dynamics after gravistimulation.Weaknesses:Genetically the relocalization of D6PK depends on the LAZY protein family, but some essential details are missing in this study. On the one hand, NGR1-GFP does not associate with the BFA compartments and maintains its association with the PM and amyloplasts. On the other hand, D6PK relies on GNOM, via vesicle trafficking sensitive to BFA, suggesting that D6PK follows a different relocalization route than NGR1 which is BFA-insensitive. Based on these observations, D6PK relocalization requires the LAZY proteins, but D6PK and NGR1 relocalize through independent routes. How can this be interpreted or reconciled?

Response: Since we demonstrated that D6PK does not relocalize in the absence of NGR proteins, we conclude that NGR1 acts upstream of D6PK. The molecular mechanism driving this interaction is not fully understood; however, it is evident that NGR1 triggers the mobilization of D6PK. Despite previous investigations into D6PK mobility, the underlying mechanisms remain elusive. Notably, despite its sensitivity to BFA, D6PK does not localize to BFA bodies and does not undergo conventional endocytosis (https://doi.org/10.1016/j.devcel.2014.05.006). We fully acknowledge the importance and interest in gaining a better understanding of these processes, and it will be a focal point of our future research.

Two other works (now published) provide valuable and fundamental findings related to the mechanism examined in the current manuscript and display complementary and similar results to the ones shown in the current manuscript. Given the similarities in the examined mechanisms, these preprints should be referenced, recognized, and discussed in the manuscript under review. It is assumed that the three projects were independently developed, but the results of these previous works should be addressed and taken into account at least during the discussion and when drawing any conclusions. This does not mean that this work is less relevant. On the contrary, some of the observations that seem to be redundant are more solid, and firm conclusions can now be drawn from them.

Response: We have included and discussed these works in the revised discussion

**Reviewer #2 (Public Review):**
Summary:This manuscript addresses what rapid molecular events underly the earliest responses after gravity-sensing via the sedimentation of starch-enriched amyloplasts in columella cells of the plant root cap. The LAZY or NEGATIVE GRAVITROPIC RESPONSE OF ROOTS (NGR) protein family is involved in this process and localizes to both the amyloplast and to the plasma membrane (PM) of columella cells.The current manuscript complements and extends Nishimura et al., Science, 2023. Kulich and colleagues describe the role of the LZY2 protein, also called NGR1, during this process, imaging its fast relocation and addressing additional novel points such as molecular mechanisms underlying NGR1 plasma membrane association as well as revealing the requirement of NGR1/LZY2, 3,4 for the polar localization of the AGCVIII D6 protein kinase at the PM of columella cells, in which NGR1/LZY2 acts redundantly with LZY3 and LZY4.The authors initially monitored relocalization of functional NGR1-GFP in columella cells of the ngr1 ngr2 ngr3 triple mutant after 180-degree reorientation of the roots. Within 10 -15 min NGR1-GFP signal disappeared from the upper PM after reorientation and reappeared at the lower PM of the reoriented cells in close proximity to the sedimented amyloplasts. Reorientation of NGR1-GFP occurred substantially faster than PIN3-GFP reorientation, at about the same time or slightly later than a rise in a calcium sensor (GCaMP3) just preceding a change in D2-Venus auxin sensor alterations. Reorientation of NGR1-GFP proved to be fast and not dependent on a brefeldin A-sensitive ARF GEF-mediated vesicle trafficking, unlike the trafficking of PIN proteins, like PIN3, or the AGCVIII D6 protein kinase. Strikingly, the PM association of NGR1-GFP was highly sensitive to pharmacological interference with sterol composition or concentration and phosphatidylinositol (4)kinase inhibition as well as dithiothreitol (DTT) treatment interfering with thioester bond formation e.g. during S-acylation. Indeed, combined mutation of a palmitoylation site and polybasic regions of NRG1 abolished its PM but not its amyloplast localization and rendered the protein non-functional during the gravitropic response, suggesting NRG1 PM localization is essential for the gravitropic response. Targeting the protein to the PM via an artificially introduced N-terminal myristoylation and an ROP2-derived polybasic region and geranylgeranylation site partially restored its functionality in the gravitropic response.Strengths:This timely work should be of broad interest to plant, cell and developmental biologists across the field as gravity sensing and signaling may well be of general interest. The point that NGR1 is rapidly responsive to gravistimulation, polarizes at the PM in the vicinity to amyloplast and that this is required for repolarization of D6 protein kinase, prior to PIN relocation is really compelling. The manuscript is generally well-written and accessible to a general readership. The figures are clear and of high quality, and the methods are sufficiently explained for reproduction of the experiments.Weaknesses:Statistical analysis has been performed for some figures but is lacking for most of the quantitative analyses in the figure legends.

Response: We added this information to the figure legends

The title claims a bit more than what is actually shown in the manuscript: While auxin response reporter alterations are monitored, "rapid redirection of auxin fluxes" are not really directly addressed and, while D6PK can activate PIN proteins in other contexts, it is not explicitly shown in the manuscript that PIN3 is a target in the context of columella cells in vivo. A title such as "Rapid redirection of D6 protein kinase during Arabidopsis root gravitropism relies on plasma membrane translocation of NGR proteins" would reflect the results better.

Response: We modified the title to Rapid translocation of NGR proteins driving polarization of PIN-activating D6 protein kinase during root gravitropism

Fig. 4: The point that D6PK is transcytosed cannot be made here based on the data of these authors. They should have used a photoswitchable version of NGR1 to show that the same molecules observed at the upper PM are translocated to the lower PM. Nishimura and colleagues actually did that for NGR4. However, this is a lot of work and maybe for NGR1 that fusion would have too low fluorescence intensity (as it was the case for NGR3). So, I think a rewording would be sufficient such as NGR-dependent reorientation of D6PK plasma membrane localization" as this does not say, from where it comes to the lower PM. Theoretically, the signal could also be amyloplast-derived or newly synthesized (or just folded) NGR1-GFP.

Response: We fully agree and rephrased the text using translocation instead of transcytosis

The authors make a model in which D6PK AGCVIII kinase-dependent on NGRs activates PIN3 to drive auxin fluxes. However, alterations in auxin responses are observed prior to PIN3 reorientation. They should explain this discrepancy better and clearly describe that this is a working hypothesis for the future rather than explicitly proven, yet.
**Reviewer #3 (Public Review):**
The mechanism controlling plant gravity sensing has fascinated researchers for centuries. It has been clear for at least the past decade that starch-filled plastids (termed statoliths) in specialised gravity-sensing columella cells sense changes in root orientation, triggering an asymmetric auxin gradient that alters root growth direction. Nevertheless, exactly how statolith movement triggers PIN auxin efflux carrier activation and auxin gradient formation has remained unclear until very recently. A series of new papers (in Science and Cell) and this manuscript report how LAZY proteins (also referred to as NEGATIVE GRAVITROPIC 50 RESPONSE OF ROOTS; NGR) play a pivotal role in regulating root gravitropism. In terms of their overall significance, their collective findings provide seminal insights into the very earliest steps for how plant roots sense gravity which are arguably the most important papers about root gravitropism in the past decade.In the current manuscript, Kulich et al initially report (through creating a functional NGR1-GFP reporter) that "NGR1-GFP displayed a highly specific columella expression, which was most prominent at the PM and the statolith periphery." Is NGR1-GFP expressed in shoot tissues? If yes, is it in starch sheath (the gravity-sensing equivalent of root columella cells)? The authors also note "NGR1-GFP signal from the PM was not evenly distributed, but rather polarized to the lower side of the columella cells in the vicinity of the sedimented statoliths (Fig. 1A)." and (when overexpressing NGR-GFP) "chloroplasts in the vicinity of the PM strongly correlated with NGR1 accumulating at the PM nearby, similar to the scenario in columella" suggesting that NGR1 does not require additional tissue-specific factors (i.e. trafficking proteins or lipids) to assist in its intracellular movement from plastid to PM.

Response: Yes, NGR1, also called LAZY2 is expressed in the inner hypocotyl tissues, according to https://doi.org/10.1104/pp.17.00942. Unfortunately, we saw very little signal with our NGR-GFP construct, possibly due to NGR1-GFP weak signal and/or NGR1 being expressed only exclusively in the inner tissues.

Next, the authors study the spatiotemporal dynamics of NGR1-GFP re-localisation with other early gravitropic signals and/or components Calcium, auxin, and PIN3. The temporal data presented in Figure 1 illustrates how the GCaMP calcium reporter (in panel E) revealed "the first signaling event in the root gravitropic bending is the statolith removal from the top membrane, rather than its arrival at the bottom" It appeared that the auxin DII-VENUS reporter was also changing rapidly (panel G) - was this detectable BEFORE statolith re-sedimentation?

Response: In our data (Figure 1G), we observe that the increase in signal at the top side begins prior to starch sedimentation, in contrast to the bottom side, where the decrease starts only after starch grains land on the bottom membrane. While this observation aligns with our hypothesis and other data, we refrained from commenting on it due to the small differences between the first 2-3 timepoints, which are obscured by noise. This phenomenon arises because the DII response relies on protein degradation and is relatively slow. Hence, for rapid tracking of the auxin response, we utilized auxin-induced calcium as a proxy, with NPA treatment serving as a negative control.

Please can the authors explain their NPA result in Fig 1E? Why would treatment with the auxin transport inhibitor NPA block Ca signalling (unless the latter was dependent on the former)?

Response: Auxin induces rapid calcium transients (e.g., http://dx.doi.org/10.1016/j.cub.2015.10.025). Consequently, when auxin reaches the bottom elongation zone approximately 5-6 minutes after rotation, we observe an increased GCaMP signal at this location. Notably, when we inhibit PIN function using NPA, the GCaMP signal persists, but the difference between the top and bottom diminishes. This validates that the calcium transients at the bottom side can be interpreted as monitoring increase in auxin accumulation as a result of auxin transport.

They go on to note "This initial auxin asymmetry is mediated by PIN-dependent auxin transport, despite visible polarization of PIN3 can be detected only later" which suggests that PIN activity was being modified prior to PIN polarisation.In contrast to other proteins involved in gravity response like RLDs and PINs, NGR1 localization and gravity-induced polarization does not undergo BFA-sensitive endocytic recycling by ARF-GEF GNOM. This makes sense given NGR1 is initially targeted to plastids, THEN the PM. Does NGR1 contain a cleavable plastid targeting signal? The authors go on to elegantly demonstrate that NGR1 PM targeting relies on palmitoylation through imaging and mutagenesis-based transgenic ngr rescue assays.

Response: Yes, there is weakly conserved plastid targeting signal on NGR1. Although we also started researching in this direction, we quickly realized, that two other groups showed very comprehensive data regarding NGR plastid localization.

Finally, the authors demonstrate that gravitropic-induced auxin gradient formation is initially dependent on PIN3 auxin efflux activation (prior to PIN3 re-localisation). This early PIN3 activation process is dependent on NGR1 re-targeting D6PK (a PIN3 activating kinase). This elegant molecular mechanism integrates all the regulatory components described in the paper into a comprehensive root gravity sensing model.
**Recommendations for the authors:**

**Reviewer #1 (Recommendations For The Authors):**
Minor comments:Line 83: This construct fully rescued the agravitropic bending phenotype of the ngr1/2/3 triple mutant (see further).What does it mean the see further in this context?

Response: It is a reference to the second part of the manuscript (Fig. 3, Supplementary Fig S3, Fig S4), where we extensively address the complementation with wild type and point mutated versions of NGR. There we show that the construct we are using is functional. This does not prove, but strongly imply that the GFP signal we obtain is relevant. We updated the text to point this out.

Line 101: Timing of events during the gravitropic responseWhen describing the equipment employed and the rotation applied to the samples, "the vertical stage microscope and minimized the time required for rotating the sample. 180{degree sign} rotation..."The authors mentioned a travel time of 5 minutes first and later of 15 minutes for the relocalization of NGR1. Are these two different experiments? Were there two different rotation angles or degrees applied? Could the authors please rephrase this part of the description to answer these questions and help the reader understand how the assay performed?

Response: We added this explanation to the text.

Figure 1 E, F, and G.Could the authors please provide pictures and/or videos for the PIN3 localization dynamics, intracellular calcium transients, and auxin reporter DII-Venus? In other words, show the complementing images for Figure 1E, 1F, and 1G as the authors did for Figure 2D where authors presented the pictures and the corresponding quantification plots.

Response: We wanted to avoid overcrowding the figure, but we would also love to show the videos. Therefore, we did additional supplementary movie 3, where we put all the additional observations.

Line 194: This implies the existence of posttranslational modifications such as S-acylation to associate with PM.Why is this specific modification suggested/examined and no other modification? What is the criteria to select this kind of modification? Based on what premises? Could the authors elaborate on that? Could the authors please include references?

Response: Thank you for this comment. We of course first checked the prediction tools which have shown very strongly conserved S-acylation side. We now clarified this in the text and added other modifications as an example. Later on, we rule out myristoylation (that happens on the glycins) and prenylation (it happens only at the C-terminus CAAX box).

Line 255: NGR1 PM localization is synergistically mediated by polybasic regions and a palmitoylation siteSimilarly to the previous commentary, How and why are these regions examined/analyzed? Likewise, why is the palmitoylation site selected? Please provide some background, criteria, and references.

Response: Here, we clearly state that the prediction of the palmitoylation site is made based on the GPS lipid prediction tool.

As for the polybasic region, these can be seen upon manual inspection of the primary protein sequence. We simply looked at the protein and saw it there. We rephrased the text so that it is more clear.

**Reviewer #2 (Recommendations For The Authors):**
Please, proofread the manuscript for style and minor language errors.Statistical analysis has been performed for some figures but is lacking for most of the quantitative analyses in the figure legends. Where it has been performed it is not given what "n" number of roots, cells, or plasma membranes were analyzed NGR1-GFP and no information is given whether the data is derived from a representative experiment or several or pooled data from several experiments. This certainly requires revision in Fig. 1D-G, Fig. 2B-D, Fig. S2 B,E, Fig. 3B,D, F-H, Fig. S.3 B,D, Fig. S. 4 ,E-H, Fig. 4 D.

Response: Thank you, we added this information to the figure legends.